# Reduced Lean Body Mass and Cardiometabolic Diseases in Adult Males with Overweight and Obesity: A Pilot Study

**DOI:** 10.3390/ijerph15122754

**Published:** 2018-12-05

**Authors:** Shirine Khazem, Leila Itani, Dima Kreidieh, Dana El Masri, Hana Tannir, Roberto Citarella, Marwan El Ghoch

**Affiliations:** 1Department of Nutrition and Dietetics, Faculty of Health Sciences, Beirut Arab University, P.O. Box 11-5020 Riad El Solh, Beirut 11072809, Lebanon; shirine_sk@live.com (S.K.); l.itani@bau.edu.lb (L.I.); d.kraydeyeh@bau.edu.lb (D.K.); dana.masri@bau.edu.lb (D.E.M.); hana.tannir@bau.edu.lb (H.T.); 2CTR Centre of Rehabilitation Therapy, Via Fratelli Cervi, 59/e, 42124 Reggio Emilia, Italy; citarella@ctr-re.it; 3Nutrition and Dietetics Program—CTR Centre of Rehabilitation Therapy, Via Fratelli Cervi, 59/e, 42124 Reggio Emilia, Italy

**Keywords:** lean body mass, obesity, type 2 diabetes, metabolic syndrome, cardiometabolic diseases

## Abstract

Little is known about the reduction in lean body mass (LBM) and its health consequences in overweight and in obesity, especially in males. Therefore, we aimed to assess the prevalence of low LBM in treatment-seeking adult males with overweight and obesity and the association with cardiometabolic diseases, i.e., type 2 diabetes, cardiovascular diseases and dyslipidemia. A body composition assessment was conducted by a bio-impedance analyzer (InBody 170) among a total of 110 males, of whom 72 were overweight and obese and were referred to the Outpatient Clinic in the Department of Nutrition and Dietetics at Beirut Arab University (BAU) in Lebanon, and 38 were normal-weight participants of similar ages. The participants with overweight and obesity were then categorized as being with or without low LBM. Of the sample of 72 participants, 50 (69.4%) met the criteria for reduced LBM and displayed a significantly higher prevalence of cardiometabolic diseases (i.e., type 2 diabetes, cardiovascular diseases and dyslipidemia) than those with normal LBM (36.0% vs. 9.1%; *p* = 0.019). Logistic regression analysis showed that low LBM increases the odds of having cardiometabolic diseases by nearly 550% (odds ratio (OR) = 5.46, 95% confidence interval (CI) = 1.31–26.39, *p* < 0.05) after adjusting for total fat and central adiposity. Treatment-seeking adult males with overweight and obesity displayed a great prevalence of reduced LBM, which seems to be strongly associated with cardiovascular and metabolic diseases.

## 1. Introduction

Extensive research has been conducted on body composition patterns in obesity [1]; however, most studies have focused on fat body mass and there is a lack of data on the reduction in lean body mass (LBM) as a primary outcome and its consequences for health among this population, especially in males [2]. This is an important limitation in the field, since, in the general population, low LBM has been shown to be associated with a higher risk of cardiovascular events [3], diabetes [4], metabolic syndrome [5], longer hospitalization [6], and mortality [7,8].

Recent, well-designed studies have underlined the association between low LBM and cardiometabolic risk factors in females with or without obesity [9,10,11]. However, these results cannot be generalized across genders and therefore should still be considered preliminary. To the best of our knowledge, there is a lack of data derived from males who are overweight or suffer obesity in this regard. Therefore, the identification of low LBM and its consequence on health outcomes in this population is of critical importance, with the application of potential therapeutic strategies (e.g., physical activity interventions, high-protein diets and protein supplements) focused on limiting LBM deterioration as a clinical priority [12].

On the other hand, it is important to note that defining low LBM in individuals with excess weight based only on lean mass (without accounting for body mass) may be misleading, because patients with overweight and obesity tend to have a relatively high lean mass [13]. Hence, the criteria for low LBM, when based solely on this parameter, may not be met among these individuals in whom the prevalence may therefore be greatly underestimated [13].

The aims of the present study were, firstly to assess the prevalence of low LBM in treatment-seeking adult males with overweight and obesity by applying three different definitions, which, in addition to appendicular lean mass (ALM), also involve body size (weight and body mass index (BMI)) [14,15,16]. The second aim was to compare which of these three definitions is more clinically useful for detecting any potential association between low LBM and cardiovascular and metabolic diseases, namely, type 2 diabetes, cardiovascular diseases and dyslipidemia in this population.

## 2. Materials and Methods

Participants were selected from a cohort of 335 patients seeking treatment at the Beirut Arab University (BAU) nutritional and weight management outpatient clinic in the Department of Nutrition and Dietetics of BAU in Beirut, Lebanon, between 2017 and 2018. Recruitment was possible following consecutive referrals to the clinic and only male patients were considered eligible. A total of 72 males out of the 111 assessed for eligibility were included because they fulfilled the inclusion criteria for being aged ≥18 years, with a BMI ≥ 25.0 kg/m^2^ and at least one weight loss-responsive co-morbidity (i.e., type 2 diabetes, cardiovascular disease, etc., two or more risk factors), as defined by the Adult Treatment Panel III [17]. Patients who were on medication that affect body weight and/or body composition or who were affected by diseases associated with weight loss or severe psychiatric disorders were excluded. The power analysis for the sample size was conducted retrospectively based on the probability of having cardiometabolic diseases when exposed/unexposed to reduced lean body mass and the corresponding odds in the study sample. Using G power post hoc computation of achieved power given an α = 0.05 and odds ratio of 5.6 from cross tabulation and assuming a moderate association with covariates (*R*^2^ = 0.25), a sample of 72 cases would give a power of 88% (having a 12% chance of rejecting H_0_ when it is true/or having a false positive) [18].

Thirty-eight normal-weight male participants (BMI ≥ 18.5 and <25 kg/m^2^) were recruited from university listservs and advertisements. All were healthy and weight-stable with no history of cardiometabolic diseases or other significant medical conditions and having no current medication intake. The study design was reviewed and approved by the Institutional Review Board of BAU (No. 2018H-0050-HS-R-0293), and all participants gave informed written consent for the anonymous use of their personal information.

A questionnaire was administered to participants with overweight and obesity and controls in order to retrieve information regarding their medical history and demographic and social conditions (age, marital status etc.).

Body weight and height were measured by a dietician affiliated with the study using a weighing scale (SECA 2730-ASTRA, Germany) and medical stadiometer respectively. Each participant’s BMI was then calculated as per the standard formula of body weight (kg) divided by height (m) squared.

A bioimpedance analyzer (InBody 170, BIOSPACE, China) was used to measure the body composition. It provides separate body mass readings for different segments of the body and uses an algorithm incorporating impedance, age and height in order to estimate total and regional body fat and fat-free mass. A dietician affiliated with the study and with information based on a clinical assessment ensured that participants were normally hydrated and had abstained from drinking (i.e., consuming caffeine or alcohol) and exercising for at least 12 h before the test. The total fat and lean mass percentages and the Appendicular Lean Mass (ALM) were calculated using standard formulas [19].

Low LBM was defined by means of three definitions as follows:(1)Batsis et al. [14]: ALM/BMI <0.789(2)Levine and Crimmins [15]: (ALM/weight) × 100% < 25.72(3)Oh et al. [16]: (ALM/weight) × 100% < 29.60

Cardiovascular and metabolic disease in this study indicates the presence of any diseases such as type 2 diabetes, cardiovascular diseases (coronary heart disease, stroke, transient ischemic attack, and peripheral arterial disease) and dyslipidemia (a decreased concentration of high-density lipoprotein cholesterol, and an increased concentration of high-density lipoprotein cholesterol and triglycerides) based on self-reported diagnosis, either simultaneously or separately.

Descriptive statistics were calculated as means, standard deviation, frequencies, and proportions. The χ^2^ test and student’s *t*-test were used to compare proportions and means, respectively, between participants with normal weight and those with overweight and obesity, and between participants with and without low LBM in the clinical sample. Simple and multiple logistic regression analyses were performed to calculate the odds of the presence of cardiometabolic diseases in the clinical sample with low LBM. The data were tested by a quantile-quantile normality plot, which revealed that variables were normally distributed. All analyses were performed using SPSS version 25.0 (IBM Corp.; IBM, Armonk, NY, USA). Statistical significance was considered at *p* < 0.05.

## 3. Results

Table 1 shows the sociodemographic characteristics of the two study sample groups that comprised of 110 males: 72 participants with overweight or obesity (mean BMI of 33.69 ± 5.85 kg/m^2^) and 38 participants with normal weight (mean BMI of 22.31 ± 1.83 kg/m^2^) (Table 1). The former group displayed a reduced LBM according to all three definitions used (ALM/BMI = 1.11 ± 0.14 vs. 0.85 ± 0.14; *p* < 0.01; (ALM/weight) × 100 = 36.09 ± 2.48 vs. 27.86 ± 3.34; *p* < 0.01) (Table 2), while the two groups were similar in age (31.26 ± 12.68 vs. 32.79 ± 13.65 years; *p* = 0.560) (Table 1). According to the three definitions from low LBM, in the overweight and obesity groups, the prevalence varied between 23.6% and 69.4%. None of the participants in the normal-weight group were affected by low LBM (Table 2).

The low LBM group had a higher prevalence of cardiometabolic diseases (36.0% vs. 9.1%; *p* = 0.019) (Table 3). Moreover, the group with low LBM, when compared with the normal group, had a significantly higher BMI, waist-to-hip ratio, total body fat percentage, visceral fat mass, and fat-free mass percentage (Table 4).

Logistic regression analysis showed that having low LBM increases the odds of having cardiovascular and metabolic diseases by nearly 555% (OR = 5.46, 95% CI = 1.31–26.39, *p* < 0.05) after adding the total and visceral fat mass to the model (Table 5).

## 4. Discussion

The present study aimed to provide preliminary data through a pilot study on the prevalence of low LBM in adult males with overweight and obesity and to assess any potential association between low LBM with cardiometabolic diseases, namely, type 2 diabetes, cardiovascular diseases and dyslipidemia, in this population. In turn, two major findings were revealed.

Firstly, in our population of men affected by overweight and obesity (across a wide age range), we found that the prevalence of low LBM was 23.6%, 30.6% and 69.4%, respectively, based on the three definitions we used [14,15,16]. These fall within the wide prevalence range of 0 to 100% reported for males [20], however, using the same formulas, other studies reported different results. For instance, Stoklossa and colleagues found a higher prevalence of low LBM: 58.8%, 47.1% and 100%, respectively. This variation may depend on the samples included. Specifically, in her study, participants with class II and III obesity (mean BMI = 44.0 ± 6.3 kg/m^2^) were included and they, logically, may have a high prevalence of low LBM. Our sample, conversely, included participants who were overweight, or of the class I and II obesity (mean BMI = 33.69 ± 5.85 kg/m^2^) categories [21]. In general, we speculate that higher prevalence tends to be in study samples accounting for body mass (i.e., weight and BMI) [20]. Furthermore, a low prevalence may also be explained by the use of definitions that have primarily been developed in the course of studies on older cohorts [20].

Second, we found that only one definition was able to detect any association between low LBM and cardiometabolic diseases [16]. Namely the definition proposed by Oh and colleagues [16] seems to be more clinically useful in detecting the association between low LBM and cardiovascular and metabolic diseases. In fact, nearly 36% of participants with low LBM had type 2 diabetes, cardiovascular disease or dyslipidemia, with these conditions strongly associated, where, low LBM increases the odds of presenting cardiometabolic diseases by more than 550% after adjusting for total body fat and central adiposity, which are known to be associated with cardiometabolic diseases.

Two clinical implications can be derived from our findings: Firstly, that an awareness of the high prevalence of reduced LBM in treatment-seeking adult males with overweight and obesity should be raised among clinicians as well as patients; secondly, our results reveal the importance of screening for low LBM in this population, since it seems to be strongly associated with cardiometabolic diseases.

Our study has certain strengths. Firstly, to the best of our knowledge, it is the first to assess low LBM in the Arab region and one of the few studies to assess low LBM among individuals with overweight and obesity, by taking into account ALM, weight and BMI [22]. In fact, as far as we are aware, only one study has used this definition to identify low LBM, in this case, among Italian treatment-seeking patients with obesity [13].

However, the study has certain limitations. First, our results need to be interpreted with caution because they may not apply to patients treated in other settings different from an outpatient clinic. Second, body composition was measured using a bio-impedance analyzer; despite being validated, it is still not accepted as the gold-standard technique for patients with overweight and obesity [23]. Third, no biochemical testing was conducted; this means that we are unable to shed any light on the mechanisms and implications of the sarcopenia that we observed in adult males with overweight and obesity. Fourth, by studying an exclusive male sample, our findings cannot be extended to females with obesity. Fifth, the cross-sectional design of our study should be considered as another limitation. Sixth, no objective measures regarding eating habits and levels of physical activity were available. Finally, in the diagnosis of cardiometabolic diseases, we relied on self-reporting data.

## 5. Conclusions

Our findings provide evidence that adult males with overweight and obesity who are seeking treatment have a high prevalence of low LBM. This condition seems to be strongly associated with the presentation of weight-related diseases, such as cardiovascular disease, type 2 diabetes and dyslipidemia. Therefore, the identification of low LBM in this population is clinically useful.

## Figures and Tables

**Table 1 ijerph-15-02754-t001:** Anthropometric and sociodemographic characteristics of the study population.

	Total Sample *N* = 110	Normal Weight *N* = 38	Overweight and Obesity *N* = 72	*p*-Value
Age (Years)	32.26 (13.28)	31.26 (12.68)	32.79 (13.65)	0.560
BMI	29.76 ± 7.28	22.31 ± 1.83	33.69 ± 5.85	*p* < 0.0001
Marital status				0.694
Unmarried	69 (63.3)	25 (65.8)	44 (62.0)	
Married	40 (36.7)	13 (34.2)	27 (38.0)	
Employment				0.446
Unemployed	43 (39.1)	13 (34.2)	30 (41.7)	
Employed	67 (60.9)	25 (65.8)	42 (58.3)	

BMI: Body mass index.

**Table 2 ijerph-15-02754-t002:** Prevalence of reduced lean body mass (LBM) among the study participants using different definitions.

Definition of Low LBM	Reference	Cut-Off Point	Low LBM					Mean Values		
			Normal Weight *N* = 38		Overweight and Obesity *N* = 72		*p*-Value	Normal Weight *N* = 38	Overweight and Obesity *N* = 72	*p*-Value
			Normal LBM	Low LBM	Normal LBM	Low LBM				
ALM/BMI	Batsis et al. [14]	<0.789	38 (100.0)	0 (0.0)	50 (69.4)	22 (30.6)	<0.01	1.11 ± 0.14	0.85 ± 0.14	<0.01
ALM/Weight × 100%	Levine and Crimmins [15]	<25.72	38 (100.0)	0 (0.0)	55 (76.4)	17 (23.6)	<0.01	36.09 ± 2.48	27.86 ± 3.34	<0.01
	Oh et al. [16]	<29.60	38 (100.0)	0 (0.0)	22 (30.6)	50 (69.4)	<0.01	36.09 ± 2.48	27.86 ± 3.34	<0.01

ALM: Appendicular Lean Mass.

**Table 3 ijerph-15-02754-t003:** Prevalence of cardiometabolic disease among the sample study.

	Total Sample *N* = 110	Normal Weight *N* = 38	Overweight and Obesity *N* = 72	*p*-Value	Overweight and Obesity *N* = 72		*p*-Value
Disease					Normal LBM	Low LBM *	
Type 2 diabetes				*p* = 0.144			*p* = 0.804
No	105 (96.3)	37 (100.0)	68 (94.4)		21 (95.5)	47 (94.0)	
Yes	4 (3.7)	0 (0.0)	4 (5.6)		1 (4.5)	3 (6.0)	
Dyslipidemia				*p* = 0.008			*p* = 0.012
No	97 (89.0)	37(100.0)	60 (83.3)		22 (100.0)	38 (76.0)	
Yes	12 (11.0)	0 (0.0)	12 (16.7)		0 (0.0)	12 (24.0)	
Cardiovascular Disease and Hypertension				*p* = 0.012			*p* = 0.093
No	98 (89.0)	37 (100.0)	61 (84.7)		21 (95.5)	40 (80.0)	
Yes	11 (10.1)	0 (0.0)	11 (15.3)		1 (4.5)	10 (20.0)	
Cardiometabolic Disease				*p* = 0.0004			*p* = 0.019
No	89 (81.7)	37 (100.0)	52 (72.2)		20 (90.9)	32 (64.0)	
Yes	20 (18.3)	0 (0.0)	20 (27.8)		2 (9.1)	18 (36.0)	

* Low LBM defined as ALM/weight × 100% < 29.6.

**Table 4 ijerph-15-02754-t004:** Anthropometric characteristics and body composition patterns by BMI status and category of low LBM.

					Overweight and Obesity		
	Total *N* = 110	Normal Weight *N* = 38	Overweight and Obesity *N* = 72	*p*-Value	Normal LBM *N* = 49	Low LBM * *N* = 23	*p*-Value
BMI	29.76 ± 7.28	22.31 ± 1.83	33.69 ± 5.85	*p* < 0.0001	28.70 ± 2.45	35.88 ± 5.57	*p* < 0.0001
Waist to hip ratio	0.92 ± 0.06	0.86 ± 0.03	0.95 ± 0.04	*p* < 0.0001	0.92 ± 0.02	0.97 ± 0.03	*p* < 0.0001
Total fat mass	26.44 ± 16.20	11.02 ± 3.40	34.58 ± 14.22	*p* < 0.0001	21.28 ± 4.98	40.43 ± 12.97	*p* < 0.0001
% Body fat	27.00 ± 10.68	16.03 ± 4.57	32.79 ± 8.11	*p* < 0.0001	23.94 ± 4.30	36.68 ± 6.06	*p* < 0.0001
Fat free mass	64.10 ± 11.75	56.83 ± 9.93	67.93 ± 10.83	*p* < 0.0001	67.18 ± 8.02	68.26 ± 11.91	*p* > 0.05
% Fat-free mass	72.54 ± 11.47	82.63 ± 10.07	67.21 ± 8.11	*p* < 0.0001	76.05 ± 4.30	63.32 ± 6.07	*p* < 0.0001
Visceral fat mass	9.77 ± 5.73	3.97 ± 1.65	12.83 ± 4.64	*p* < 0.0001	8.23 ± 2.05	14.86 ± 3.96	*p* < 0.0001
Total body water	47.19 ± 8.02	42.39 ± 5.69	49.72 ± 7.94	*p* < 0.0001	49.15 ± 5.83	49.97 ± 8.75	*p* > 0.05
ALM/weight × 100%	30.70 ± 4.98	36.09 ± 2.48	27.86 ± 3.34	*p* < 0.0001	31.65 ± 1.66	26.19 ± 2.37	*p* < 0.0001
ALM/weight × 100% < 29.6				*p* < 0.0001			
Normal LBM	60 (54.5)	38 (100)	22 (30.6)				
Low LBM	50 (45.5)	0 (0.0)	50 (69.4)				

* Low LBM defined as ALM/weight × 100% < 29.6.

**Table 5 ijerph-15-02754-t005:** Odds of cardiometabolic diseases with reduced LBM among participants with overweight and obesity.

	ALM/BMI < 0.789	ALM/weight × 100% < 25.72	ALM/weight × 100% < 29.6
Cardio-Metabolic Diseases	Odds (95% CI)		
LBM			
No low LBM	1.00	1.00	1.00
Low LBM	0.998 (0.840–1.185)	0.969 (0.816–1.150)	5.463 (1.131–26.399)
Percent visceral fat from total fat	2.204 (0.656–7.408)	1.421 (0.377–5.359)	0.976 (0.832–1.145)

CI: Confidence interval.

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
