# Peer review of "Reduced Lean Body Mass and Cardiometabolic Diseases in Adult Males with Overweight and Obesity: A Pilot Study"

_ijerph, 2018, doi:10.3390/ijerph15122754_

Reviewer 1 Report

The present study proposes to assess the prevalence of low LBM in treatment-seeking adult males with overweight and obesity and the association with cardiometabolic diseases, i.e., type 2 diabetes, cardiovascular diseases, and dyslipidemia.

In general, the study is well delineated; the statistical analysis seems to be appropriate, although the presentation of the results demand substantial improvement. On the other hand, the proposed topic of the present study has already well characterized in the literature (for example, a quick search in the NCBI database, Pubmed presents 8269 articles). In this sense, the introduction session does not clarify the relevance of the study and what advances are proposed for the area of knowledge. Still, in this question, the objective of the study must be presented more clearly, that is, what it was intended to be; to identify the prevalence of LBM in obese individuals, or to compare the best criterion for the evaluation of LBM.

In the methods session, taking into account the technical limitations in the analysis of bioimpedance, there is no description of the procedures adopted for the control of fluid intake and hydration level of the patient/subject of the study. In the report of the statistical analysis, there is no information about the data passed by the analysis of normality.

In the results session, there is no general information about the patients (casuistry). Besides, Table 2 is very confusing, and there is information that is not required to be displayed, such as the value of Chi-Square (could be reported as a supplement).

Finally, the discussion session does not compare the results to the facts that are already well established in the literature and does not discuss the main advances. Also, as the main objectives are not clear, and in this way, it is difficult to understand what the main issues/outcomes to be discussed. However, the study's limitations paragraph is adequate.

Author Response

Reviewer 1

In general, the study is well delineated; the statistical analysis seems to be appropriate, although the presentation of the results demand substantial improvement. On the other hand, the proposed topic of the present study has already well characterized in the literature (for example, a quick search in the NCBI database, PubMed presents 8269 articles). In this sense, the introduction session does not clarify the relevance of the study and what advances are proposed for the area of knowledge. Still, in this question, the objective of the study must be presented more clearly, that is, what it was intended to be; to identify the prevalence of LBM in obese individuals, or to compare the best criterion for the evaluation of LBM.

Response: Now we clarify in the Introduction section the issues raised by the reviewer (Lines 61-64 and 67-71 and 80 and 83).  

In the methods session, taking into account the technical limitations in the analysis of bioimpedance, there is no description of the procedures adopted for the control of fluid intake and hydration level of the patient/subject of the study. In the report of the statistical analysis, there is no information about the data passed by the analysis of normality.

Response: Done as suggested, now we mentioned the procedure adopted in which we controlled fluid intake and hydration level of participants (Lines 119-121).

Moreover the data was tested by Q-Q normality plot, which revealed that variables were normally distributed (Check the figures below). This statement was added in the Statistical analysis section (Lines 137-138).

In the results session, there is no general information about the patients (casuistry). Besides, Table 2 is very confusing, and there is information that is not required to be displayed, such as the value of Chi-Square (could be reported as a supplement).

Response: Due to the Reviewer 2 request, we added Table 1 on general information about participants. As suggested we remove Chi-Square values from table 2 to simplify.

Finally, the discussion session does not compare the results to the facts that are already well established in the literature and does not discuss the main advances. Also, as the main objectives are not clear, and in this way, it is difficult to understand what the main issues/outcomes to be discussed. However, the study's limitations paragraph is adequate.

Response: We thank the reviewer for the valuable comment, now we discuss our results with previous well-established literature where this was possible (Lines 174-180).

Moderate English changes required

Response: Done as suggested

Reviewer 2 Report

-GENERAL COMMENTS: The topic is one of importance given the high number of adults showing overweight and obesity. If conducted with academic rigor, this article has the potential to be of value for practitioners and policymakers around the prevalence, cost and prevention of complications for this problems. Furthermore, in my opinion the topic and premise of the study would sit well within the journal to which it was submitted. The authors should be commended for undertaking this study, however, there are a major concerns with the manuscript that require attention prior to publication. These will be discussed below relative to the sections of the manuscript.

-TITLE: The title is correct as it reflects correctly the objective of the work.

-SUMMARY: Correct

-INTRODUCTION

This section did not provide a clear rationale for carrying out the study (for example, why is your research question important? What gap in the literature is the study addressing?. I suggest in this section should be improved, with more information related with importance of the reduced lean body mass and cardiometabolic diseases in adult males with overweight and obesity. The authors consider that tetrapolar bioelectrical impedance is the best method to evaluate lean body mass? Also, please describe the hypothesis in this section.

-MATERIAL AND METHODS

In this section, you need to clearly describe how individuals were approached, how many were approached, how many were eligible, consented or refused may be recommended in order to improve the quality of the manuscript.

Likewise, more detail about information calculate sample size and data should be provided. For example, explain why odds of 5.6 with a power of 80% and 5% significance?

Although the design and methods are described elsewhere in detail, it does not mean that authors must provide few basics at least, specifically, from the body weight, height, BIA analysis must be provided in more detail (technical error, professional who take the measurement, etc).

-RESULTS

Please, include table with medical history and demographic and social conditions (age, marital status etc.).

- DISCUSION

Does these three definitions is more clinically useful for detecting any potential association between low LBM and cardiovascular and metabolic diseases, namely, type 2 diabetes, cardiovascular diseases and dyslipidaemia, in this population tissue to skeletal muscle ?  Is these three definitions more valuable than using muscle mass (kg, %) for example?  How does the ROC compare to the other proposed measures in the literature?

Author Response

Reviewer 2

INTRODUCTION

This section did not provide a clear rationale for carrying out the study (for example, why is your research question important? What gap in the literature is the study addressing? I suggest in this section should be improved, with more information related with importance of the reduced lean body mass and cardiometabolic diseases in adult males with overweight and obesity. The authors consider that tetrapolar bioelectrical impedance is the best method to evaluate lean body mass? Also, please describe the hypothesis in this section.

Response: We improved the introduction as suggested (Lines 61-64 and 67-71 and 80 and 83). The tetrapolar bioelectrical impedance is the not considered the best method to evaluate lean body mass, and we mentioned its limitations in the Discussion section (Lines 204-206). 

MATERIAL AND METHODS

In this section, you need to clearly describe how individuals were approached, how many were approached, how many were eligible, consented or refused may be recommended in order to improve the quality of the manuscript.

Response: Done as suggested describing selection, eligibility and inclusion criteria (Lines 87-91).

Likewise, more detail about information calculate sample size and data should be provided. For example, explain why odds of 5.6 with a power of 80% and 5% significance?

Response: Done as suggested, now more details about sample size calculations are provided (Lines 96-102).

Although the design and methods are described elsewhere in detail, it does not mean that authors must provide few basics at least, specifically, from the body weight, height, BIA analysis must be provided in more detail (technical error, professional who take the measurement, etc).

Response: Done as suggested (Lines 112-114 and 119-121). 

RESULTS

Please, include table with medical history and demographic and social conditions (age, marital status etc.).

Response: We added Table 1 with the available information.

DISCUSION

Does these three definitions is more clinically useful for detecting any potential association between low LBM and cardiovascular and metabolic diseases, namely, type 2 diabetes, cardiovascular diseases and dyslipidaemia, in this population tissue to skeletal muscle? Is these three definitions more valuable than using muscle mass (kg, %) for example?  How does the ROC compare to the other proposed measures in the literature?

Response: We thank the reviewer for the comment. Only one definition was able to detect any association between low LBM and cardiometabolic diseases. Namely the definition proposed by Oh and colleagues seems to be more clinically useful for detecting the association between low LBM and cardiovascular and metabolic diseases. This statement was added in the Discussion section (Lines 185-187).

On the other hand, it is important to note that defining low LBM in individuals with excess weight, based only on lean mass (kg,%) without accounting for body mass may not misleading, because patients with overweight and obesity tend to have a relatively high lean mass, and this statement has been already mentioned in the Introduction section (Line 75-77).

Finally ROC analysis was not conducted since there is no gold standard for diagnosis of low LBM. Moreover, in spite of the significant finding we cannot claim that the measure is a gold standard.

Round  2

Reviewer 1 Report

In general, the changes made significantly improved the quality of the study and the clarity in the transmission of the main results. However, the manuscript should be analyzed by a native of the English language.

Reviewer 2 Report

The additional statistics and clarification around the methods and limitations has improved the manuscript.